# Individual and community-level factors associated with discrimination among women aged 15–49 years in Bangladesh: Evidence based on multiple indicator cluster survey

**Iqramul Haq[1]\*, Md. Mizanur Rahman Sarker[1], Sharanon Chakma[2]**

1 Department of Agricultural Statistics, Sher-e-Bangla Agricultural University, Dhaka, Bangladesh,
2 Department of Development and Poverty Studies, Sher-e-Bangla Agricultural University, Dhaka, Bangladesh

\* iqramul.haq@sau.edu.bd

**Data Availability Statement:** https://mics.unicef.org/.

## Abstract

### Background

This research aimed to examine the factors at both the individual and community levels that are linked to discrimination experienced by women aged 15–49 in Bangladesh.

### Methods

The relevant data was taken from the 2019 Multiple Indicator Cluster Survey in Bangladesh. The risk factors for discrimination against women in Bangladesh were determined using multilevel logistic regression models.

### Results

The overall prevalence of discrimination against women was found to be 10.4% (95% CI: 10.1–10.6). Based on the final model (Model 1V), at the individual level higher odds of discrimination were observed among women from poor (AOR:1.21,95%CI: 1.12–1.32) and middle income households (AOR:1.12, 95%CI:1.02–1.22) compared to those from rich households etc. Women who have never used ICT were 1.27 times (AOR = 1.27, 95% CI = 1.07–1.51) higher odds of discrimination when compared with women who were ICT exposed. Respondents who married before 18 years 10% more likely to (AOR = 1.10, 95% CI:1.02–1.19) discriminated than women married aged 18 years old or above. Women from urban communities were 15% less likely to experience discrimination than their rural counterparts. In comparison to the Sylhet Division, women in the Barisal, Chattogram, Dhaka, Khulna Mymensingh, Rajshahi, and Rangpur Divisions were respectively 3.02, 1.84, 1.68, 2.06, 4.97, 4.06, and 1.74 times more likely to experience discrimination.

### Conclusion

Findings revealed that various individual-level factors such as wealth index, CEB, ICT exposure, marital status, functional difficulty, age, women's happiness, magazine and radio

**Funding:** The author(s) received no specific funding for this work.

**Competing interests:** The authors have declared that no competing interests exist.

exposure, age at marriage, current contraceptive use, polygamy, husband beating, place of attack, and household head age were found to have a significant association with women discrimination. Community-level factors such as residence and division were also found to have a notable impact on discrimination. Policymakers should incorporate substantial components targeting both individual and community levels into intervention programs with the goal of raising awareness about women's discrimination.

## Introduction

Discrimination against women violates human rights principles, hindering their equal participation in politics, social interactions, the economy, and culture. This hinders progress and challenges women's capabilities and meaningful contributions to humanity. Approximately 2.4 billion women of working age face unequal access to the economy, while 178 nations maintain legal barriers that hinder their full participation in economic activities [1]. Almost half of the women who responded in a 2019 American Economic Association (AEA) survey said that compared to just 3% of the male respondents, they had experienced discrimination based on sex [2]. Around 100 million women in Asia are believed to be 'missing' as a result of discriminatory practices in terms of healthcare and nutrition access, prenatal sex selection, or even outright neglect [3]. The Association of Southeast Asian Nations (ASEAN) reports that the COVID-19 pandemic disproportionately affects women and girls, hindering gender equality and escalating violence [4]. While there has been significant progress in women's participation and inclusion in traditionally male-dominated fields in recent decades, the level of men's engagement in traditionally female-dominated domains has remained relatively stagnant [5]. Discrimination is often influenced by cultural norms, traditions, religion, geography, and other factors [6]. Women belonging to Native American, Black, and Latina backgrounds are more likely to report experiencing gender-based discrimination in various areas, including healthcare, compared to white women [7]. Discrimination patterns differ for married and unmarried women, with married women facing higher levels of discrimination in areas such as mobility, property, employment, and education, while unmarried women tend to report more discrimination in mobility and behavior [8]. Colleagues who wear hijabs encounter formal discrimination, such as receiving fewer callbacks for job applications or facing obstacles in completing application processes, as well as experiencing interpersonal discrimination, such as perceived hostility and reduced expectations of receiving job offers [9, 10]. Madurese women, particularly those from disadvantaged backgrounds, encounter cultural and systemic pressures that hinder gender equality [11].

If we look in contemporary Bangladesh, sociocultural values and conventions play a significant role on the issue of women discriminations. Based on the 2016 data on labor force participation, it was found that 81.9% of men and 35.6% of women in Bangladesh were employed [12]. In terms of higher education, there is a significant gender disparity at the university level. In 2001, only 24.3% of all students at public universities in Bangladesh were female, while male enrollment exceeded that of females by more than three times, accounting for 75.7% [13]. At the national level, the proportion of employed men and women has increased from 67.5 and 15.2% in 2004 to 68.3% and 22.9 percent, respectively, indicating discrimination in employment [13].

For instance, in the Muslim community, the birth of a boy is greatly celebrated, while the entrance of a girl is not ritually celebrated by the family and the society [14, 15]. The birth of sons is eagerly anticipated in Bangladesh, as they are expected to become the primary earners for the family and provide financial support to their parents in their old age. On the other hand, daughters are often seen as a burden to the family [16, 17]. Upon getting married,

daughters in Bangladesh often experience emotional and physical detachment from their parental household. Unfortunately, their contributions to the family are seldom recognized or valued [18]. Furthermore, these occupations are often seen as less valuable in terms of the workforce, as women are often confined to lower-level positions with narrower job responsibilities and lesser financial incentives [19]. Female workers don't always get the right job postings, timely pay raises, or promotions based on their qualifications and competencies, despite doing the same work [20]. Apart from that, it becomes more difficult for working women even to keep their occupations due to cultural hurdles, gender discrimination, social danger, physical challenges, and a lack of family support [19]. Socio-cultural and patriarchal norms contribute to the perpetuation of gender discrimination and practices that grant men more control over resources while systematically restricting women's access to opportunities [21]. In Bangladesh, the intersection of physical disability and being female presents numerous intricate obstacles to inclusion in the country's socioeconomic and cultural aspects [22]. There have been limited studies conducted in Bangladesh on women's discrimination, and most of the studies applied binary logistic regression. In addition, there are some studies conducted in many countries using the probit model [23–25]. However, none of the previous studies attempted to investigate the factors influencing discrimination against women at both the individual and community levels using a multilevel analysis approach. That is why we applied a multilevel model to identify both individual and community-level factors that influence women discrimination in Bangladesh. This research will assist policymakers in identifying the factors contributing to discrimination against women at both individual and community levels in Bangladesh. The findings will be valuable for the government to align its national goals with international objectives like SDG 5 (Gender Equality) and SDG 10 (Reduced Inequalities), in order to address and mitigate discrimination effectively.

## Conceptual framework

Women discrimination has been influenced by a range of characteristics, including factors such as age [23–27], marital status [28–30], age of marriage, functional difficulties [31, 32], household age, wealth index [30, 33], currently pregnant, currently contraceptive use, child born, husband beating, place of attack, happiness status, magazine exposure, radio exposure, TV exposure, ICT). Additionally, community characteristics such as area and region [34–36] also play a role in women discrimination.

Fig 1 visually presents a conceptual model depicting the dual-level process of women discrimination.

1. Level 1: Individual levels, which include women characteristics and household characteristics.

2. Level 2: Community level

It is worth highlighting that these two levels are interconnected and possess a multilevel nature. This implies that women's individual characteristics, along with household characteristics, are intertwined with community-level factors.

## Methodology

### Data source and design

The analysis based on the secondary data from nationwide representative survey named 2019 Multiple indicator cluster survey which is funded by UNICEF [37]. In the MICS 2019, two

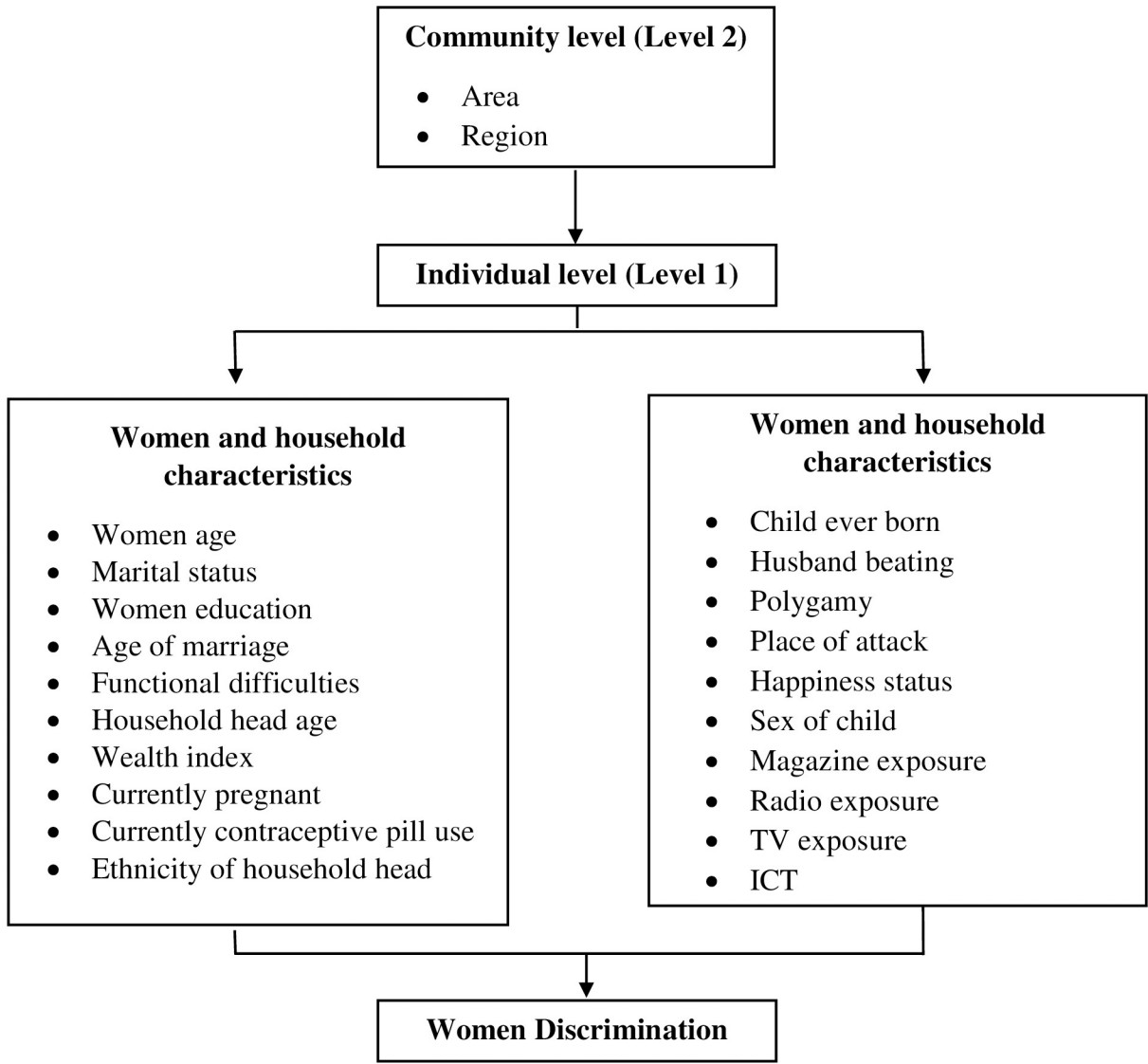

**Fig 1. Conceptual framework of women discrimination with hierarchical structure.**

stage stratified sampling is applied. The urban and rural areas of each district were chosen as the primary strata for sampling. At the first stage, 3,220 enumeration areas within each stratum were chosen with probability proportionate to size. A systematic sample of 20 households was selected from each PSU (Primary Sampling Unit). A total of 68,711 women between the ages of 15 and 49 were interviewed in the 64,000 households. The analysis included ever-married and currently married women aged 15–49, therefore this analysis is based on 64,378 women.

## Outcome variable

In this analysis the response variable is discrimination against women. In Bangladesh, women between the ages of 15 and 49 reported experiencing personal discrimination or harassment within the past 12 months. These incidents were based on various factors, including ethnic or immigration origin, sex, sexual orientation, age, religion or belief, disability, and other unspecified reasons [37].

For each case, a value '1' is assigned if a respondent felt discriminated against or harassed for the above mentioned reason and '0' is considered as women did not feel discriminated. Then adding the values of these seven variables (0–7) and thus outcome variable is categorized as "Yes" to any one of these questions constituted the women felt discriminated and otherwise categorized "No". Therefore, the outcome variable is defined as:

$$Discrimination = \begin{cases} \textbf{Yes}, & \textbf{If women felt discriminated due to any one mentioned reasos} \\ \textbf{No}, & \textbf{Otherwise} \end{cases}$$

## Independent variables

A set of covariates related to women's feelings of discrimination were considered for analysis. All the selected covariates are classified as individual-level factors and community-level factors. At the individual level, this study considered the following: Women's age (in years) (15–19, 20–34, and 35–49), women's education (no education, primary, secondary and higher), marital status (Currently married, formerly married and never married), happiness (happy, neutral, unhappy), household head age (15–21, 22–34, 35–90), age at marriage (<18 and 18+), ethnicity of household head (Bengali and others), functional difficulty (yes, no), child death (0, 1, and 2 +), child born (0, 1–2 and 3+), child survive (0, 1,2, and 3+), magazine exposure (yes, no), radio exposure (yes, no), TV exposure (yes, no), husband beating (yes, no), place of attack (home and outside home), sex of child (male, female), more housewives (yes, no), ICT (not exposed, exposed), currently pregnant (yes, no) and wealth status (poor, middle, and rich). The wealth index is a composite measure of wealth that is created using principal component analysis. Based on their assets and the final factor ratings, each household is given a wealth score. The purpose of the wealth index is to rank households based on their wealth (from poorest to richest) without providing information on absolute poverty, income, or expenditure levels, since accurately measuring those can be difficult. The significance of the wealth index lies in its impact on household health, its usefulness in targeting poverty alleviation programs, and its ability to be cross-tabulated with other variables in the dataset. In the specific case of Bangladesh, the Bangladesh Wealth Index incorporates 25 variables from the Bangladesh MICS 2019 survey [37]. The poor category was formed by merging the poorest and poorer groups, while the rich category was created by combining the richest and richer segments of the study participants. Additional information about the development of the wealth index is available in [38, 39]. In this study, a community was defined as a census enumeration cluster or block. The type of residence (urban or rural) was added as a proxy measure to take into consideration community characteristics. The division variable was also taken into account, and divisions like Barishal, Chattogram, Dhaka, Khulna, Mymensingh, Rajshahi, Rangpur, and Sylhet were included.

## Statistical analysis

To describe the study population's key socio-demographic characteristics, including both individual and community-level elements, descriptive statistics were used. These statistics provide a summary and description of the data, allowing for a better understanding of the study population's characteristics.

In bivariate analysis, the chi-square test of significance was applied in the bivariate setup to examine the relation between response and covariates. Multilevel logistic regression models are commonly used in research studies to address the hierarchical structure of data and analyze the impact of individual and group-level factors on an outcome variable. This modeling approach enables the simultaneous consideration of individual-level and contextual-level

effects, leading to a more comprehensive understanding of the factors associated with the outcome [40, 41]. The data set has a community-level influence. Instead of using a single-level modeling strategy, one could think of using multilevel modeling to examine this kind of data [42]. It should be emphasized that a two-stage stratified sampling approach was used to get the data for this investigation [37]. As a result, there is a hierarchy of levels at which reliance between observations occurs [42, 43]. Given the hierarchical structure of the MICS 2019 data, multilevel logistic regression was employed to identify the effects of individual and community-level factors on discrimination against women. This analysis involved utilizing a series of four models.

Before applying any multilevel model, the intraclass correlation coefficient (ICC) should be determined and defined by

$$ICC = \frac{\sigma^2_{cluster}}{\left(\sigma^2_{cluster} + \frac{\pi^2}{3}\right)};$$

where $\frac{\pi^2}{3} \approx 3.29$ refers to the standard logistic distribution. If the value of intra-class correlation (ICC) >0, a multilevel logistic regression model can be utilized [44]. The proportional change in variance (PCV) was calculated for Model II, III, and IV in relation to the variance observed in the null model. This calculation was performed to demonstrate the explanatory power of the factors included in the models for discrimination against women. The PCV was determined using the formula PCV = $\left(\frac{V_n - V_{mi}}{V_n}\right) \times 100$, where $V_n$ and $V_{mi}$ represent the variance obtained in the null model and the variance in each successive model, respectively. In essence, the PCV was obtained by subtracting the variance of each model from the variance observed in the null model [45].

First, we only fitted a random intercept to an empty model (Model I). The second model, which incorporated each individual factor, was then fitted (Model II). In Model III, only community-level factors were included. However, in the final model (Model IV), both individual and community-level variables were incorporated. The adjusted odds ratios (AOR) and their corresponding 95% confidence intervals were presented as fixed effects for each model. The model fit was evaluated using Akaike's Information Criterion (AIC), Bayesian Information Criterion (BIC), and deviance. The current model exhibited better fit compared to the previous model, as indicated by lower values of AIC, BIC, and deviance. Numerous studies have recognized multicollinearity as a high degree of dependency between independent variables. If the tolerance number was 0.1 or below and the variance inflation factor (VIF) cutoff point of 5 or 10 was taken into consideration, it would be cause for multicollinearity.

Statistical Package for the Social Sciences (SPSS) version 25 and R version 4.0.0 were used for the management of data and analysis.

## Results

### Sociodemographic and economic characteristics of women

According to Table 1, approximately 47.6% of the women were aged between 20 and 34 years. Over half of the women (61.5%) had received secondary and higher education, while 22.7% had completed primary education and 15.8% had no formal education. Furthermore, among women with available data, a higher proportion was observed among those who did not have any functional difficulties (96.9%), had given birth to 1–2 children (44.6%), had no child deaths (90.2%), had 2 surviving children (28.3%), and got married before the age of 18 (64.7%). In terms of household composition, the majority of households were Bengali (98.8%). Additionally, among the heads of households, 67.0% fell within the age range of 35–90 years.

**Table 1. Background characteristics of women.**

| Variables | Frequency | Percentage |
|---|---:|---:|
| **Individual Level Factor** | | |
| **Age** | | |
| 15–19 | 11950 | 18.6 |
| 20–34 | 30658 | 47.6 |
| 35–49 | 21770 | 33.8 |
| **Women education** | | |
| No education | 10187 | 15.8 |
| Primary | 14615 | 22.7 |
| Secondary and higher | 39577 | 61.5 |
| **Functional difficulties (age 18–49 years)** | | |
| Yes | 1760 | 3.1 |
| No | 55886 | 96.9 |
| **Age of marriage** | | |
| <18 | 34801 | 64.7 |
| 18+ | 18915 | 35.2 |
| **More housewives (Polygamy)** | | |
| Yes | 1610 | 2.5 |
| No | 62768 | 97.5 |
| **Ethnicity of household head** | | |
| Bengali | 63626 | 98.8 |
| Others | 752 | 1.2 |
| **Household head age** | | |
| 15–21 | 718 | 1.4 |
| 22–34 | 16159 | 31.6 |
| 35–90 | 34195 | 67.0 |
| **Wealth index** | | |
| Poor | 23595 | 36.7 |
| Middle | 12988 | 20.2 |
| Rich | 27795 | 43.2 |
| **Currently pregnant** | | |
| Yes | 2722 | 5.3 |
| No | 48400 | 94.7 |
| **Currently contraceptive use** | | |
| Yes | 32054 | 66.3 |
| No | 16346 | 33.7 |
| **Child born** | | |
| 0 | 15958 | 24.8 |
| 1–2 | 28696 | 44.6 |
| 3+ | 19724 | 30.6 |
| **Sex of newborn** | | |
| Male | 26359 | 54.4 |
| Female | 22061 | 45.6 |
| **Child survives** | | |
| 0 | 16203 | 25.2 |
| 1 | 12298 | 19.1 |
| 2 | 18222 | 28.3 |
| 3+ | 17655 | 27.4 |

(*Continued*)

**Table 1.** (Continued)

| Variables | Frequency | Percentage |
|---|---:|---:|
| **Child death** | | |
| 0 | 58060 | 90.2 |
| 1 | 5167 | 8.0 |
| 2+ | 1151 | 1.8 |
| **Husband beating** | | |
| Yes | 15541 | 25.0 |
| No | 46643 | 75.0 |
| **Place of attack** | | |
| Home | 2313 | 88.3 |
| Outside home | 308 | 11.7 |
| **Marital status of woman** | | |
| Currently married | 51121 | 79.4 |
| Formerly married | 2594 | 4.0 |
| Never married | 10662 | 16.6 |
| **Happiness** | | |
| Happy | 54464 | 84.6 |
| Neutral | 7140 | 11.1 |
| Unhappy | 2775 | 4.3 |
| **Magazine exposure** | | |
| No | 58852 | 91.4 |
| Yes | 5526 | 8.6 |
| **Radio exposure** | | |
| No | 62706 | 97.4 |
| Yes | 1672 | 2.6 |
| **TV exposure** | | |
| No | 20279 | 31.5 |
| Yes | 44099 | 68.5 |
| **Information and communications technology (ICT)** | | |
| Not exposed | 1403 | 2.2 |
| Exposed | 62975 | 97.8 |
| | | **Community Level Factor** |
| **Residence** | | |
| Urban | 15094 | 23.4 |
| Rural | 49284 | 76.6 |
| **Division** | | |
| Barishal | 3465 | 5.4 |
| Chattogram | 12514 | 19.4 |
| Dhaka | 16316 | 25.3 |
| Khulna | 7578 | 11.8 |
| Mymensingh | 4181 | 6.5 |
| Rajshahi | 8521 | 13.2 |
| Rangpur | 7081 | 11 |
| Sylhet | 4722 | 7.3 |

The highest percentage of households were classified as rich based on the wealth index (43.2%), followed by poor (36.7%), and middle (20.2%). Table 1 further reveals that a significant majority of women were not currently pregnant (94.7%), and employed a method to

Table 2. Prevalence of women discrimination.

| Variable | N | % | 95% CI |
|---|---|---|---|
| **Women felt discrimination** | | | |
| Yes | 6694 | 10.4 | 10.1–10.6 |
| No | 57542 | 89.6 | 88.4–91.2 |

prevent pregnancy (66.3%). The majority of newborns were male (54.4%). It was found that a significant proportion of women experienced domestic violence within their homes (88.3%), while a smaller percentage faced physical abuse from their husbands (25.0%). Moreover, most women were currently married or in a union (79.4%). Most of the women were happy (84.6%). Most of them did not read magazines (91.4%). Table 1 further illustrates that most of them did not have a radio (97.4%). Majority of them had a TV (68.5%). Most of them were exposed to information and communications technology (ICT) (97.8%). Table 1 demonstrates that the highest percentage of participants in this study were from the Dhaka division (25.3%) followed by Chattogram (19.4%), Rajshahi (13.2%) and Khulna (11.8%) respectively and the lowest percentage of participants were from the Barishal division (5.4%). The majority of participants live in the rural area (76.6%).

## Prevalence of discrimination against women in Bangladesh

According to the analysis, the overall prevalence of discrimination against women in Bangladesh was determined to be 10.4% (95% CI: 10.1–10.6) (Table 2).

## Relationship between discrimination and individual and community variables

Table 3 evidently shows the association between percentage distribution and discrimination, which was accompanied by sociodemographic characteristics. The study identified several factors that were significantly associated (p-value < 0.001 or p-value < 0.01) with discrimination against women. These factors include individual-level variables such as age, functional difficulty, age at marriage, presence of polygamy (more housewives), household head age, wealth index, current pregnancy status, contraceptive use, number of children ever born, number of surviving children, husband's history of beating, place of attack, marital status, women's happiness, exposure to magazines, radio, TV, and information and communications technology (ICT). Additionally, community-level factors such as residence type and division were also found to be significant factors associated with discrimination against women. Subsequently, it was also observed that the rate of discrimination decreases with increasing women's age and wealth index. For example women aged 15–19 years (13.9%) mostly experienced discrimination compared to 35–49 years (8.6%). Women from poor (12.7%) households were mostly discriminated than rich (8.4%). Similarly there was a significant negatively association household head age and discrimination. Women having household heads aged 15–21 years (14.1%) were mostly discriminated compared to women having household heads aged 35–90 years (8.4%).

In this study, the discrimination against women also higher among functional difficulty women (15.4%), early marriage (9.7%), husband have more housewives (17.2%), women beat by husband (13.7%), currently pregnant women (9.1%), women have no children (15.5%), women have no children survive (15.4%), outside home attack (34.2%), and unhappy women (22.9%). Discrimination against women was high among women who read magazines (12.3%), women who listened to radio (14.5%), those who did not watch television (10.9%), and those who were not exposed to media and information and communication networks

**Table 3. Association between selected covariates and discrimination against women in Bangladesh.**

| Variables | Discrimination | | |
|---|---|---|---|
| | Yes (%) | No (%) | $\chi^2$ value (p-value) |
| **Individual level factors** | | | |
| **Age (in years)** | | | |
| 15–19 | 13.9 | 86.1 | 237.166(<0.001) |
| 20–34 | 10.4 | 89.6 | |
| 35–49 | 8.6 | 91.4 | |
| **Women education** | | | 3.799 (0.150) |
| No education | 10.0 | 90.0 | |
| Primary | 10.7 | 89.3 | |
| Secondary and higher | 10.4 | 89.6 | |
| **Functional difficulties (FD)** | | | 61.512(<0.001) |
| Yes | 15.4 | 84.6 | |
| No | 9.7 | 90.3 | |
| **Age of marriage** | | | 26.183(<0.001) |
| <18 | 9.7 | 90.3 | |
| 18+ | 8.3 | 91.7 | |
| **More housewives (Polygamy)** | | | 80.541(<0.001) |
| Yes | 17.2 | 82.8 | |
| No | 10.2 | 89.8 | |
| **Ethnicity of household head (HH)** | | | 0.074 (0.786) |
| Bengali | 10.4 | 89.6 | |
| Others | 10.1 | 89.9 | |
| **Household head (HH) age** | | | 44.037(<0.001) |
| 15–21 | 14.1 | 85.9 | |
| 22–34 | 9.6 | 90.4 | |
| 35–90 | 8.4 | 91.6 | |
| **Wealth index** | | | 252.504(<0.001) |
| Poor | 12.7 | 87.3 | |
| Middle | 10.8 | 89.2 | |
| Rich | 8.4 | 91.6 | |
| **Currently pregnant** | | | 26.722(<0.001) |
| Yes | 9.1 | 90.9 | |
| No | 8.8 | 91.2 | |
| **Currently contraceptive use** | | | 17.356(<0.001) |
| Yes | 8.4 | 91.6 | |
| No | 9.6 | 90.4 | |
| **Child born** | | | 593.053(<0.001) |
| 0 | 15.5 | 84.5 | |
| 1–2 | 9.2 | 90.8 | |
| 3+ | 8.1 | 91.9 | |
| **Sex of newborn** | | | 0.039 (0.844) |
| Male | 8.8 | 91.2 | |
| Female | 8.7 | 91.3 | |

(*Continued*)

**Table 3.** (Continued)

| Variables | Discrimination | | |
|---|---|---|---|
| **Child survives** | | | 603.980 ($< .001$) |
| 0 | 15.4 | 84.6 | |
| 1 | 9.9 | 90.1 | |
| 2 | 8.9 | 91.1 | |
| 3+ | 7.8 | 92.2 | |
| **Child death** | | | 3.480 (0.176) |
| 0 | 10.5 | 89.5 | |
| 1 | 9.7 | 90.3 | |
| 2+ | 10.0 | 90.0 | |
| **Husband beating** | | | 289.287 ($<0.001$) |
| Yes | 13.7 | 86.3 | |
| No | 9.0 | 91.0 | |
| **Place of attack** | | | 14.914 ($<0.001$) |
| Home | 24.0 | 76.0 | |
| Outside home | 34.2 | 65.8 | |
| **Marital status of woman** | | | 673.908 ($<0.001$) |
| Currently married | 8.8 | 91.2 | |
| Formerly married | 16.2 | 83.8 | |
| Never married | 16.6 | 83.4 | |
| **Happiness status** | | | 728.084 ($<0.001$) |
| Happy | 9.2 | 90.8 | |
| Neutral | 15.3 | 84.7 | |
| Unhappy | 22.9 | 77.1 | |
| **Magazine exposure** | | | 21.841 ($<0.001$) |
| No | 10.2 | 89.8 | |
| Yes | 12.3 | 87.7 | |
| **Radio exposure** | | | 31.190 ($<0.001$) |
| No | 10.3 | 89.7 | |
| Yes | 14.5 | 85.5 | |
| **TV exposure** | | | 7.093 ($<0.01$) |
| No | 10.9 | 89.1 | |
| Yes | 10.2 | 89.8 | |
| **ICT** | | | 53.385 ($<0.001$) |
| Not exposed | 16.4 | 83.6 | |
| Exposed | 10.3 | 89.7 | |
| **Community level** | | | |
| **Residence** | | | 42.794 ($<0.001$) |
| Urban | 9.0 | 91.0 | |
| Rural | 10.9 | 89.1 | |
| **Division** | | | 972.728 ($<0.001$) |
| Barishal | 12.3 | 87.7 | |
| Chattogram | 8.3 | 91.7 | |
| Dhaka | 7.9 | 92.1 | |
| Khulna | 10.4 | 89.6 | |
| Mymensingh | 20.4 | 79.6 | |
| Rajshahi | 15.6 | 84.4 | |
| Rangpur | 9.8 | 90.2 | |
| Sylhet | 6.0 | 94.0 | |

(16.4%). Women in rural areas (10.9%) experienced more discrimination than in urban areas (9.0%). According to Table 3, the highest proportion of women facing discrimination was observed in the Mymensingh division (20.4%), followed by Rajshahi (15.6%), Barishal (12.3%), and Khulna (10.4%) divisions, respectively. While the lowest percentage of women experiencing discrimination was found in the Sylhet division (6.0%).

## Determinants of discrimination against women

Four models of multilevel logistic regression were fitted in the analysis (Table 4). Based on the VIF values falling below 1.5 and the tolerance limit being greater than 0.1, there was no evidence of multicollinearity in the current study.

## Random effects (measures of variation) and model comparison

**Null model (Models I).** As indicated in Model I (null model) of Table 4, there were significant variations in the likelihood of experiencing discrimination among different clusters or communities. According to the provided table, the ICC for the null model is 0.213. This suggests that approximately 21.3% of the variations in discrimination against women can be attributed to differences across clusters or communities. Based on the table, 20.9%, 17.3%, and 15.9% are the respective ICC values for models II, III, and IV. PCV values for models II, III, and IV are 2.25%, 22.5%, and 30.3%, respectively. Moreover, the higher PCV value of 0.303 in the model IV suggests that approximately 30.3% of the variation in discrimination among women can be attributed to factors at both the individual and community levels.

## Fixed effect results

**Individual-level model (Model II).** In this model II, we included individual-level covariates as fixed effects. These covariates include the wealth index, number of children ever born, ICT exposure, marital status of the woman, functional difficulties (age 18–49 years), age, happiness, magazine exposure, radio exposure, TV exposure, age of marriage, current pregnancy status, current use of contraception, more housewives status, husband's history of beating, place of attack, and age of the household head.

**Community-level model (model III).** According to model III in Table 4, women residing in urban areas were 16% (AOR = 0.84, 95% CI 0.75–0.94) less likely to face discrimination compared to women from rural areas. Women in the Mymensingh division were 4.45 times (AOR = 4.45, 95% CI: 3.51–5.64) more discriminated than women in the Sylhet division, while women in Rajshahi were 3.82 times (AOR = 3.82, 95% CI: 3.08–4.74) more, Barishal was 2.68 times (AOR = 2.86, 95% CI: 2.10–3.44) more, Khulna were 2.00 times (AOR = 2.00, 95% CI: 1.60–2.49) more, Chattogram were 96% (AOR = 1.96, 95% CI: 1.37–2.10) more, Rangpur were 81% (AOR = 1.81, 95%, CI: 1.44–2.26) more and Dhaka were 58% (AOR = 1.58, 95%, CI: 1.28–1.95) more likely to experience discrimination. All of them were significantly more likely to be discriminated than women from Sylhet division.

**Individual- and community-level model (Models IV).** Women aged 20–34 years had a 13% higher likelihood of experiencing discrimination than women aged 35–49 years (AOR = 1.13, 95% CI: 1.04–1.23). Women who were currently married were 49% less likely to experience discrimination compared to women who had never married, while formerly married women were 23% less likely to face discrimination compared to the same reference category. Women who married before the age of 18 were 10% more likely to experience discrimination than those who married at age 18 or older (AOR = 1.10, 95% CI: 1.02–1.19). In model IV, the results demonstrated that those women who had functional difficulties had significantly greater odds of facing discrimination (AOR = 1.31, 95% CI: 1.12–1.53) compared to

**Table 4. Effect of women discrimination by background characteristics in Bangladesh, obtained from multilevel logistic modeling approach.**

| Variables | Model I | Model II | Model III | Model IV |
|---|---|---|---|---|
| | AOR (95% CI) | AOR (95% CI) | AOR (95% CI) | AOR (95% CI) |
| **Individual level factors** | | | | |
| **Fixed effect** | | | | |
| Intercept | 0.08[a] (0.08–0.09) | 0.22[a] (0.17–0.29) | 0.04[a] (0.04–0.05) | 0.12[a] (0.09–0.16) |
| **Age (in years)** | | | | |
| 15–19 | | 0.88 (0.77–1.00) | | 0.90 (0.79–1.02) |
| 20–34 | | 1.11[c] (1.02–1.21) | | 1.13[b] (1.04–1.23) |
| 35–49 (ref.) | | 1 | | 1 |
| **Marital status** | | | | |
| Currently | | 0.52[a] (0.45–0.60) | | 0.51[a] (0.44–0.59) |
| Formerly | | 0.78[b] (0.65–0.93) | | 0.77[b] (0.65–0.92) |
| Never (ref.) | | 1 | | 1 |
| **Age of marriage** | | | | |
| <18 | | 1.15[a] (1.07–1.24) | | 1.10[b] (1.02–1.19) |
| 18+ (ref.) | | 1 | | 1 |
| **FD** | | | | |
| Yes | | 1.34[a] (1.15–1.57) | | 1.31[b] (1.12–1.53) |
| No (ref.) | | 1 | | 1 |
| **More housewives** | | | | |
| Yes | | 1.62[a](1.39–1.90) | | 1.62[a](1.39–1.90) |
| No (ref.) | | 1 | | 1 |
| **HH age (in years)** | | | | |
| 15–21 | | 1.40[c] (1.07–1.82) | | 1.37[c] (1.05–1.78) |
| 22–34 | | 1.06 (0.97–1.16) | | 1.05 (0.96–1.15) |
| 35–90 (ref.) | | 1 | | 1 |
| **Wealth index** | | | | |
| Poor | | 1.35[a] (1.24–1.46) | | 1.21[a] (1.12–1.32) |
| Middle | | 1.20[a] (1.10–1.30) | | 1.12[c] (1.02–1.22) |
| Rich(ref.) | | 1 | | 1 |
| **Currently pregnant** | | | | |
| Yes | | 0.87 (0.75–1.02) | | 0.87 (0.74–1.02) |
| No (ref.) | | 1 | | 1 |
| **Currently contraceptive use** | | | | |
| Yes | | 0.89[b] (0.75–1.02) | | 0.88[b] (0.82–0.95) |
| No (ref.) | | 1 | | 1 |
| **Child born** | | | | |
| 0 | | 2.02[a] (1.77–2.31) | | 1.94[a] (1.69–2.22) |
| 1–2 | | 1.20[a] (1.11–1.31) | | 1.18[a] (1.08–1.28) |
| 3+(ref.) | | 1 | | 1 |
| **Husband beating** | | | | |
| Yes | | 1.44[a] (1.34–1.54) | | 1.47[a] (1.37–1.57) |
| No (ref.) | | 1 | | 1 |
| **Place of attack** | | | | |
| Home | | 3.04[a] (2.69–3.43) | | 3.05[a] (2.70–3.44) |
| Outside home(ref.) | | 1 | | 1 |

(*Continued*)

**Table 4.** (Continued)

| Variables | Model I | Model II | Model III | Model IV |
|---|---|---|---|---|
| | AOR (95% CI) | AOR (95% CI) | AOR (95% CI) | AOR (95% CI) |
| **Happiness status** | | | | |
| Happy | | 0.41[a] (0.37–0.47) | | 0.41[a] (0.36–0.46) |
| Neutral | | 0.68[a] (0.60–0.77) | | 0.69[a] (0.60–0.78) |
| Unhappy (ref.) | | 1 | | 1 |
| **Magazine exposure** | | | | |
| No | | 0.83[b] (0.74–0.92) | | 0.83[b] (0.74–0.92) |
| Yes (ref.) | | 1 | | 1 |
| **Radio exposure** | | | | |
| No | | 0.80[b] (0.68–0.94) | | 0.81[c] (0.69–0.96) |
| Yes (ref.) | | 1 | | 1 |
| **TV exposure** | | | | |
| No | | 0.93[c] (0.86–1.00) | | 0.94 (0.88–1.01) |
| Yes (ref.) | | 1 | | 1 |
| **ICT** | | | | |
| Not exposed | | 1.28[b] (1.08–1.52) | | 1.27[b] (1.07–1.51) |
| Exposed (ref.) | | 1 | | 1 |
| **Community-level factors** | | | | |
| **Area** | | | | |
| Urban | | | 0.84[b] (0.75–0.94) | 0.85[b] (0.76–0.96) |
| Rural (ref.) | | | 1 | 1 |
| **Division** | | | | |
| Barishal | | | 2.68[a] (2.10–3.44) | 3.02[a] (2.35–3.89) |
| Chattogram | | | 1.96[a] (1.37–2.10) | 1.84[a] (1.48–2.28) |
| Dhaka | | | 1.58[a] (1.28–1.95) | 1.68[a] (1.35–2.07) |
| Khulna | | | 2.00[a] (1.60–2.49) | 2.06[a] (1.64–2.59) |
| Mymensingh | | | 4.45[a] (3.51–5.64) | 4.97[a] (3.90–6.33) |
| Rajshahi | | | 3.82[a] (3.08–4.74) | 4.06[a] (3.26–5.07) |
| Rangpur | | | 1.81[a] (1.44–2.26) | 1.74[a] (1.38–2.20) |
| Sylhet (ref.) | | | 1 | 1 |
| **Random effects** | | | | |
| ICC (%) | 21.3% | 20.9% | 17.3% | 15.9% |
| PCV (%) | NA | 2.25% | 22.5% | 30.3% |
| **Model fitness** | | | | |
| AIC | 38615.6 | 36703.1 | 38290.3 | 36369.5 |
| BIC | 38633.8 | 36929.9 | 38381.0 | 36668.9 |
| Deviance | 38611.6 | 36653.1 | 38270.3 | 36303.5 |

ref. = Reference Category; Statistical Significance: [c] $p < 0.05$; [b] $p < 0.01$; [a] $p < 0.001$

women without functional difficulties. Women with husbands who had multiple wives were more likely to experience discrimination than those whose partners did not (AOR = 1.62, 95% CI: 1.39–1.90). Women whose household head was aged 15–21 years had 37% higher odds of experiencing discrimination compared to those whose household head was aged 35–90 years (AOR = 1.37, 95% CI: 1.05–1.78). The results demonstrated that women from less wealth households were more likely to face discrimination than women from rich households. For example, women from poor households were 21% (AOR = 1.21, 95% CI: 1.12–1.32) more likely discriminated than women from rich households.

Women who presently use a method to avoid pregnancy were 12% less likely to experience discrimination compared to their counterparts. Women who did not have any children had a 1.94 times higher likelihood of experiencing discrimination compared to women who had three or more children (AOR = 1.94, 95% CI: 1.69–2.22). Women who experienced physical abuse from partners were 1.47 times more likely to face discrimination than women who did not experience such abuse (AOR = 1.47, 95% CI: 1.37–1.57). Women who were victims of attacks within their own homes faced 3.05 times higher odds of experiencing discrimination compared to those who were attacked outside of their homes (AOR = 3.05, 95% CI: 2.70–3.44). Happy women had a significantly lower likelihood of experiencing discrimination, being 59% less likely (AOR = 0.41, 95% CI: 0.36–0.46) than those who reported being unhappy. Women who did not read magazines had 17% lower odds of experiencing discrimination (AOR = 0.83, 95% CI: 0.74–0.92) compared to those who did read magazines. Similarly, women who did not listen to the radio had a 19% lower likelihood of experiencing discrimination compared to those who listened to the radio. Regarding ICT exposure, women who had access to the ICT had 1.27 times greater odds of discrimination than women who had no access to the ICT.

In community level, women residing in urban areas had 15% lower odds of experiencing discrimination compared to those residing in rural areas. The results of model IV revealed significant variations in the odds of discrimination among different divisions. Women residing in the Mymensingh division had 4.97 times greater odds of experiencing discrimination compared to women in the Sylhet division (AOR = 4.97, 95% CI: 3.90–6.33). Similarly, Rajshahi, Barishal, Khulna, Chattogram, Rangpur, and Dhaka Divisions had 4.06 times, 3.02 times, 2.06 times, 1.84 times, 1.74 times, and 1.68 times higher odds, respectively, of experiencing discrimination compared to the Sylhet Division.

## Discussion

The study used multiple models to investigate the factors influencing discrimination among women in Bangladesh. The null model (Model I), which accounted for 21% of the variation, showed significant variations in discrimination between clusters or communities. Individual-level covariates were introduced into Model II as fixed effects, and it was demonstrated that these variables consistently affected the risk of discriminating, with an estimated community-level variance of 21%. Community-level variables were incorporated in Model III, indicating differences in discriminating between various regions and urban-rural divisions. Model IV included individual and community-level variables, demonstrating that person-level variables continued to play a significant role in determining prejudice. Numerous factors have been found to be strongly associated with discrimination, including wealth index, marital status, functional difficulties, age, happiness, media consumption, pregnant status, and household head age. These findings highlight the complex interplay of individual and community level factors in shaping the experience of discrimination among women in Bangladesh.

Women living in poverty in Kenya face a range of difficulties linked to traditional cultures and norms prevailing in a society that prioritizes men [30]. Additionally, the International Labor Organization (ILO) discovered that impoverished women, who often find themselves confined to their homes, lack access to training facilities [33]. This study reveals a similar result, where women who were poor faced the highest discrimination compared to the middle and rich. Moreover, poor women have a lack of access to resources compared to wealthy women.

This study also finds that women who did not give birth to any children faced the highest discrimination. A previous investigation revealed that women who are unable to conceive

experience societal stigma and internalized negative perceptions [46]. This type of discrimination is based on outdated beliefs and cultural norms, which often prevent barren women from achieving their full potential or finding fulfillment in life.

Unmarried women have been discriminated against and stereotyped in society. Unmarried women report a greater perception of discrimination in terms of mobility and behavior [28]. In Indonesia, Ukraine, and Kenya, many unmarried women frequently encounter pressures, inequalities, and discriminatory treatment [29, 30, 47]. We also find in this study that women who were never married faced the highest discrimination.

Discrimination against women with functional difficulties is a common occurrence in modern society. This study also reveals that women who have functional difficulties face the highest discrimination. Previously conducted studies found that disabled women faced discrimination in all aspects of life. Women with disabilities face intersecting forms of discrimination [31, 48, 49]. In Afghan society, women with disabilities are commonly viewed as burdensome to their families [32].

Women of all ages face challenges in their lives. The findings of this study identify that woman who aged 20–34 faced the highest discrimination. The previous studies showed that 25–44 aged faced discrimination [26, 27].

Many women who have been married young face discrimination from their families, friends, and even society. The result of this study finds that women who get married early faced the highest discrimination, which is similar to a previous study result [50].

Furthermore, the findings of this study show that residence is a characteristic that suggests that women are more likely to experience discrimination in rural areas than in urban areas. Earlier studies have also revealed that women in rural areas of Beijing encounter higher levels of discrimination compared to their counterparts in urban areas [35, 36]. Similarly, a study conducted in Bangladesh demonstrated that individuals residing in rural regions experienced a greater sense of discrimination compared to those in urban areas [34].

The multistage level model also reveals that women who are housewives, who are not happy, who are not exposed in the ICT, who read magazines, listen to music, watch television, who were not currently pregnant and did not use any method to avoid pregnancy, who faced husband beating and attack outside the home, who had 15–21 aged household head faced the highest discrimination. The findings from this study show that women from Mymensingh division faced the highest discrimination and the lowest percentage of women from Sylhet division were discriminated among eight divisions of Bangladesh.

There were both limitations and strengths to this investigation. Among the limitations of the study, one is that it did not assess important variables known to contribute to discrimination against women, such as religion and political belief. Additionally, since this study relied on secondary data, it was not possible to establish a cause-and-effect relationship between the independent and dependent variables. However, one notable aspect of this study is its novelty in presenting a current perspective on the individual and community-level factors influencing discrimination against women in Bangladesh. Furthermore, the study employed robust and advanced analytical methods, such as the two-level logistic model, to analyze the data. The primary strength of this study lies in its use of nationally representative data, with a substantial sample size of 64,378 participants. This national survey-based approach has the potential to provide valuable insights for policymakers and different stakeholder in devising intervention strategies at both the individual and community levels.

## Conclusion

In summary, this study identified factors at both the individual and community levels that are associated with discrimination against women in Bangladesh. Our findings indicate that

various individual-level factors, such as wealth status, exposure to ICT (Information and Communication Technology), functional difficulty, age of women, number of children ever born, age at marriage, husband's age, polygamy (husband having multiple wives), incidents of husband beating, place of attack, marital status, access to magazines and radio, and current contraceptive use, have a significant impact on discrimination against women. For instance, our analysis revealed that women from low- and middle-income families, those who have never used ICT, women with functional difficulties, women aged 20 to 34 years, women with fewer children, women who married before the age of 18, women whose husbands are younger (aged 15 to 21 years), those with husbands who have multiple wives and engage in domestic violence, as well as women who experience attacks at home, exhibited higher odds of experiencing discrimination compared to their counterparts. On the contrary, our findings demonstrate that women who are currently married or formerly married, women who have not been exposed to media sources such as magazines and radio, and women who are currently using contraceptive methods exhibit decreased odds of experiencing discrimination. There was a significant association between discrimination against women and variables at the community level, such as place of residence and division. Specifically, women residing in urban areas demonstrated a lower likelihood of experiencing discrimination compared to their counterparts in rural areas. Policymakers can employ a range of strategies to potentially reduce gender discrimination. These include initiatives such as increasing access to media and ICT, raising awareness about the negative consequences of early marriage and polygamy, and providing financial assistance to poor families, particularly in rural areas within the Barishal, Dhaka, Chattogram, Khulna, Mymensingh, Rajshahi, and Rangpur divisions of Bangladesh. Additionally, policymakers can enforce laws against women's discrimination throughout the entire country, promoting an environment of equality and fairness.

## Author Contributions

**Conceptualization:** Iqramul Haq.

**Data curation:** Iqramul Haq.

**Formal analysis:** Iqramul Haq.

**Methodology:** Iqramul Haq, Md. Mizanur Rahman Sarker.

**Writing – original draft:** Iqramul Haq, Md. Mizanur Rahman Sarker, Sharanon Chakma.

**Writing – review & editing:** Iqramul Haq, Md. Mizanur Rahman Sarker, Sharanon Chakma.

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
