## [Decision Letter · Decision Letter 0]

23 May 2023

PONE-D-23-03956Assessing a multilevel model of women discrimination in Bangladesh: Evidence based on multiple indicator cluster surveyPLOS ONE

Dear Dr. Haq,

Thank you for submitting your manuscript to PLOS ONE. After careful consideration, we feel that it has merit but does not fully meet PLOS ONE’s publication criteria as it currently stands. Therefore, we invite you to submit a revised version of the manuscript that addresses the points raised during the review process.

Following changes are recommenced from reviewers that need to be incorporated:

Reviewer 01:

General comment : The authors are justified to use the approach of statistical analysis opted in this paper, however, the paper is not coherent. There are lots of gaps and standalone statements through out the paper which makes it difficult to appreciate the merits of the paper. There paper requires a professional English editor to correct grammar, accuracy and completeness of some of the statements

Title:

The title of the manuscript is not aligned to the objective of the manuscript. Technically, it is expected that the authors will assess multiple models and eventually report the parsimonious multilevel model, however, this study, sought to investigate the individual and community level factors associated with discrimination among women aged 15-49 years in Bangladesh. Therefore, the author should improve the title accordingly.

Abstract

1.Under Results, change “In the bivariate setup, all the selected covariates except women's education, the ethnicity of the household head, child death, and child sex were found to be significant for discrimination (p <0.05).change statement to read, In the bivariate models, the ethnicity of the household head, child death, and child sex were significantly associated with discrimination.

2. Since the findings included individual and community level factors, they should be reported as such. i.e At the individual level higher odds of discrimination were observed among women from poor (AOR:1.21,95%CI: 1.12-1.32) and middle income households (AOR:1.12 ,95%CI:1.02-1.22) compared to those from richest households etc.

3. The findings should be reported along with their reference group so that it easy for the reader to get complete meaning of the findings on the abstract alone.

4. I suggest the author delete the sentence “Model IV in the multilevel logistic regression model was the best model based on the principles of AIC, BIC, and deviance” instead start the sentence as, Based on the final model (Model1v), at the individual level higher odds of discrimination were observed among women from poor (AOR:1.21,95%CI: 1.12-1.32) and middle income households (AOR:1.12 ,95%CI:1.02-1.22) compared to those from rich households etc

5. Under results there is a statement that reads” Women from the other seven

divisions faced more discrimination than women from the Sylhet division” The actual categories of the variable should be stated along with their results and reference category so that the results are meaningful

Conclusion: The conclusion of the abstract doesn't add much value for this study. “According to the findings of this study, policymakers should focus on individual

and community-level factors that reduce women's discrimination in Bangladesh”. The conclusion should first establish the findings that were found to be significant before the recommendation. Similarly the recommendation is not correct, the research design is cross sectional hence the factors cannot reduce discrimination.

Introduction

1. Even though, I appreciate the background provided by the researcher however in some parts of the writing the author is simply listing statements that are not synthesized carefully to ensure the coherence with the title and the concepts of discrimination. For example, the author was tempted to discuss some of the indicators of discrimination in length such as discrimination on sex, employment, and culture while some were not discussed. It is important that the author first define or discuss what entails discrimination as defined in a standard policy document to focus the study. This will guide the researcher on how to keep the rational of the study coherent with the title and objectives as well as the methodology.

2. A majority of the studies cited are outdated, doesn't provide the prevailing picture of the problem at a global level or local level.

3. Please avoid using statement such as” There is none study conducted in Bangladesh on women discrimination and most of the study applied binary logistic regression “. Instead, I suggest you use, there are limited studies..

3. The rational for the use of a multilevel model as opposed to single level is not clear i.e “That is why we applied a multilevel model to identify both individual and community-level factors that influence women discrimination in Bangladesh. We build this research by examining several socioeconomic, socio-demographic and socio-cultural factors focusing on the general issue of individual perception of women discrimination. We investigate many individual differences that can differential predict reported gender discrimination in women in the present study, as it has been observed that both self-protective and situational factors affect how prejudicial occurrences are interpreted, with a corresponding appropriate methodology”.

Materials and Methods

1. Under the outcome variable subsection, authors should reference Bangladesh MICS to validate the measurement of the outcome variable.

2. Under the covariate subsection it an academic principle to either have a conceptual framework that depict the factors that are possibly associated with the outcome (discrimination) or cite studies that used similar factors to predict discrimination. However, in this study it is difficult to assess if some of the variables used can be used to understand discrimination.

3. Please provide additional information if the author calculated the household wealth index or was already calculated in the dataset. Either way, details on how it was calculated should be provided.

4. Under statistical analysis the authors make no mention of descriptive statistics, yet it is expected the prevalence of discrimination will be reported as well as the sample distribution. Please improve the section accordingly.

5. Even though the structure of the dataset is accurately described by the authors , most of the justification remain an opinion, no literature is cited to rationalize the use of the multilevel model i.e paragraph one under statistical analysis states “

It should be emphasized that a two-stage stratified sampling approach was used to get the data for this investigation. As a result, there is a hierarchy of levels at which reliance between observations occurs. The data set has a community-level influence. Instead of using a single-level modelling strategy, one could think of using multilevel modeling to examine this kind of data”.

6. Under Statistical analysis only Bivariate and multivariate analyses techniques are cited as approaches for the analysis in this study. However, there are results on sample distribution (see table 1) which are are not accommodated by the two analysis approaches. Please improve accordingly.

Results

1. Table 1: It may be good to list the variables according to the classification i.e list individual variables first and then community level.

2. The author may consider making the table in excel for good presentation.

3. You may consider having sub sections under the results

4. Just before the results of table 2, there should be a sub section: the relationship between discrimination and individual and community variables

5. There are missing results of the overall prevalence of discrimination that must be presented along with confidence intervals

6. The results in table 2 on the association between selected covariates and discrimination against women in Bangladesh are interpreted incorrectly. The results are row percent yet they are interpreted as column percent which is not correct.

7. The Wealth index is a proxy of the standard of living for households not the women, that is why household amenities are used to derive the household wealth index. Therefore improve the statement/interpretation of the results for the multilevel model, (table 3). Please add “The household wealth index showed a significant relationship with discrimination. The results demonstrated that women from less wealth households were more likely to face discrimination than women from rich households. For example, poor women from poor households were 21% (OR = 1.21, 95% CI: 1.12-1.32) more likely discriminated than women from rich households”.

8. Generally some interpretation requires editing for grammar to ensure they make sense to the reader. The entire section need to be improved. Some of the glaring interpretations:

• The odds of discrimination were 94% higher (OR=1.94, 95%, CI: 1.69-2.22) in women who has no child than in woman with 3+ child.

• Women not exposed to information and communication network were 27% more (OR = 1.27, 95% CI=1.07-1.51) discriminated than those were exposed.

• Women currently married were 49% less (OR = 0:51, 95% CI: 0.44-0.59), and women formerly married were 23% less (OR = 0.77, 95% CI: 0.65-0.92) faced discrimination than the reference category women never married.

• The rate of discrimination was significantly 13% (OR = 1.13, 95% CI: 1.04-1.23) more for 20-34 years old women compare to those were 35-49 years old. This is not a rate instead ratios. So technically the interpretation is not correct

• According to model IV in Table 3, happy women were 59% (OR = 0.41, 95% CI: 0.36-0.46) less likely to face discrimination than those who were unhappy.

• Women who did not read magazines were 17% less likely to be discriminated (OR = 0.83, 95% CI 0.74-0.92) less likely to be discriminated than those who did not read magazines.

9. Instead of using OR for model 3 for the fixed effects, use AOR

10. Please provide the PVC-proportion change in variance for each model.

11. Please format table 3.

Discussion

1. The Discussion section need major attention. The author was just listing the factors that were found to be associated with discrimination without a detailed synthesis of the findings. Moreover, The discussion chapter mostly identifies how findings are supported or support previous studies. I suggest the author also include studies that deviate from the extant literature.

2. I suggest you delete the first paragraph under Discussion “This study collects the necessary information from the secondary data set named 2019 multiple indicator cluster (MICS) in Bangladesh. The multilevel logistic regression model is used in this study. Although there are some earlier researches in the literature that deal with a mainly comparable issue and identify gender discrimination in Bangladesh, the number of studies on women discrimination in Bangladesh is quite low. The current analysis is to uncover potential factors associated with women discrimination based on multilevel logistic analysis.

3. Please improve this sentence “It is observed from the previous study that poor women faced most discrimination compared to wealthy women mostly in every country in the world [18, 22]”

4. The strengths and limitations paragraph need further improvement. The author mentions that the study has several limitations but only mentions one when there are several limitation inherent in this study. Also the strengths of this study are not coming out clearly . For example if there are limited studies that applied multilevel modelling in Bangladesh, how is that a strength?

Conclusion

1. The entire section on conclusion must be reworked. There are lots of hanging statements and repetitions of justifications that have been emphasized in other sections. The conclusion must help the reader understand why your research should matter to them after they have finished reading the paper.

References

1. The referencing format is not presented in an intellectual manner.

Reviewer 02

The manuscript is a technically sound piece of scientific research with data that supports the conclusion. As A reviewer, I suggest author to addd more detailed description in this piece of research as statiscal analysis might not be understood by variety of readers. There are also some sweeping generalized statements which could be modified through standard acadamic English or some more reference should be added to the paragraph of refrence 10-12. Overall quality of the writing needs to be improved.

We look forward to receiving your revised manuscript.

Kind regards,

Sadia Jabeen, Ph.D.

Academic Editor

PLOS ONE

Journal Requirements:

Additional Editor Comments:

Following changes are recommenced from reviewers that need to be incorporated:

Reviewer 01:

General comment : The authors are justified to use the approach of statistical analysis opted in this paper, however, the paper is not coherent. There are lots of gaps and standalone statements through out the paper which makes it difficult to appreciate the merits of the paper. There paper requires a professional English editor to correct grammar, accuracy and completeness of some of the statements

Title:

The title of the manuscript is not aligned to the objective of the manuscript. Technically, it is expected that the authors will assess multiple models and eventually report the parsimonious multilevel model, however, this study, sought to investigate the individual and community level factors associated with discrimination among women aged 15-49 years in Bangladesh. Therefore, the author should improve the title accordingly.

Abstract

1.Under Results, change “In the bivariate setup, all the selected covariates except women's education, the ethnicity of the household head, child death, and child sex were found to be significant for discrimination (p <0.05).change statement to read, In the bivariate models, the ethnicity of the household head, child death, and child sex were significantly associated with discrimination.

2. Since the findings included individual and community level factors, they should be reported as such. i.e At the individual level higher odds of discrimination were observed among women from poor (AOR:1.21,95%CI: 1.12-1.32) and middle income households (AOR:1.12 ,95%CI:1.02-1.22) compared to those from richest households etc.

3. The findings should be reported along with their reference group so that it easy for the reader to get complete meaning of the findings on the abstract alone.

4. I suggest the author delete the sentence “Model IV in the multilevel logistic regression model was the best model based on the principles of AIC, BIC, and deviance” instead start the sentence as, Based on the final model (Model1v), at the individual level higher odds of discrimination were observed among women from poor (AOR:1.21,95%CI: 1.12-1.32) and middle income households (AOR:1.12 ,95%CI:1.02-1.22) compared to those from rich households etc

5. Under results there is a statement that reads” Women from the other seven

divisions faced more discrimination than women from the Sylhet division” The actual categories of the variable should be stated along with their results and reference category so that the results are meaningful

Conclusion: The conclusion of the abstract doesn't add much value for this study. “According to the findings of this study, policymakers should focus on individual

and community-level factors that reduce women's discrimination in Bangladesh”. The conclusion should first establish the findings that were found to be significant before the recommendation. Similarly the recommendation is not correct, the research design is cross sectional hence the factors cannot reduce discrimination.

Introduction

1. Even though, I appreciate the background provided by the researcher however in some parts of the writing the author is simply listing statements that are not synthesized carefully to ensure the coherence with the title and the concepts of discrimination. For example, the author was tempted to discuss some of the indicators of discrimination in length such as discrimination on sex, employment, and culture while some were not discussed. It is important that the author first define or discuss what entails discrimination as defined in a standard policy document to focus the study. This will guide the researcher on how to keep the rational of the study coherent with the title and objectives as well as the methodology.

2. A majority of the studies cited are outdated, doesn't provide the prevailing picture of the problem at a global level or local level.

3. Please avoid using statement such as” There is none study conducted in Bangladesh on women discrimination and most of the study applied binary logistic regression “. Instead, I suggest you use, there are limited studies..

3. The rational for the use of a multilevel model as opposed to single level is not clear i.e “That is why we applied a multilevel model to identify both individual and community-level factors that influence women discrimination in Bangladesh. We build this research by examining several socioeconomic, socio-demographic and socio-cultural factors focusing on the general issue of individual perception of women discrimination. We investigate many individual differences that can differential predict reported gender discrimination in women in the present study, as it has been observed that both self-protective and situational factors affect how prejudicial occurrences are interpreted, with a corresponding appropriate methodology”.

Materials and Methods

1. Under the outcome variable subsection, authors should reference Bangladesh MICS to validate the measurement of the outcome variable.

2. Under the covariate subsection it an academic principle to either have a conceptual framework that depict the factors that are possibly associated with the outcome (discrimination) or cite studies that used similar factors to predict discrimination. However, in this study it is difficult to assess if some of the variables used can be used to understand discrimination.

3. Please provide additional information if the author calculated the household wealth index or was already calculated in the dataset. Either way, details on how it was calculated should be provided.

4. Under statistical analysis the authors make no mention of descriptive statistics, yet it is expected the prevalence of discrimination will be reported as well as the sample distribution. Please improve the section accordingly.

5. Even though the structure of the dataset is accurately described by the authors , most of the justification remain an opinion, no literature is cited to rationalize the use of the multilevel model i.e paragraph one under statistical analysis states “

It should be emphasized that a two-stage stratified sampling approach was used to get the data for this investigation. As a result, there is a hierarchy of levels at which reliance between observations occurs. The data set has a community-level influence. Instead of using a single-level modelling strategy, one could think of using multilevel modeling to examine this kind of data”.

6. Under Statistical analysis only Bivariate and multivariate analyses techniques are cited as approaches for the analysis in this study. However, there are results on sample distribution (see table 1) which are are not accommodated by the two analysis approaches. Please improve accordingly.

Results

1. Table 1: It may be good to list the variables according to the classification i.e list individual variables first and then community level.

2. The author may consider making the table in excel for good presentation.

3. You may consider having sub sections under the results

4. Just before the results of table 2, there should be a sub section: the relationship between discrimination and individual and community variables

5. There are missing results of the overall prevalence of discrimination that must be presented along with confidence intervals

6. The results in table 2 on the association between selected covariates and discrimination against women in Bangladesh are interpreted incorrectly. The results are row percent yet they are interpreted as column percent which is not correct.

7. The Wealth index is a proxy of the standard of living for households not the women, that is why household amenities are used to derive the household wealth index. Therefore improve the statement/interpretation of the results for the multilevel model, (table 3). Please add “The household wealth index showed a significant relationship with discrimination. The results demonstrated that women from less wealth households were more likely to face discrimination than women from rich households. For example, poor women from poor households were 21% (OR = 1.21, 95% CI: 1.12-1.32) more likely discriminated than women from rich households”.

8. Generally some interpretation requires editing for grammar to ensure they make sense to the reader. The entire section need to be improved. Some of the glaring interpretations:

• The odds of discrimination were 94% higher (OR=1.94, 95%, CI: 1.69-2.22) in women who has no child than in woman with 3+ child.

• Women not exposed to information and communication network were 27% more (OR = 1.27, 95% CI=1.07-1.51) discriminated than those were exposed.

• Women currently married were 49% less (OR = 0:51, 95% CI: 0.44-0.59), and women formerly married were 23% less (OR = 0.77, 95% CI: 0.65-0.92) faced discrimination than the reference category women never married.

• The rate of discrimination was significantly 13% (OR = 1.13, 95% CI: 1.04-1.23) more for 20-34 years old women compare to those were 35-49 years old. This is not a rate instead ratios. So technically the interpretation is not correct

• According to model IV in Table 3, happy women were 59% (OR = 0.41, 95% CI: 0.36-0.46) less likely to face discrimination than those who were unhappy.

• Women who did not read magazines were 17% less likely to be discriminated (OR = 0.83, 95% CI 0.74-0.92) less likely to be discriminated than those who did not read magazines.

9. Instead of using OR for model 3 for the fixed effects, use AOR

10. Please provide the PVC-proportion change in variance for each model.

11. Please format table 3.

Discussion

1. The Discussion section need major attention. The author was just listing the factors that were found to be associated with discrimination without a detailed synthesis of the findings. Moreover, The discussion chapter mostly identifies how findings are supported or support previous studies. I suggest the author also include studies that deviate from the extant literature.

2. I suggest you delete the first paragraph under Discussion “This study collects the necessary information from the secondary data set named 2019 multiple indicator cluster (MICS) in Bangladesh. The multilevel logistic regression model is used in this study. Although there are some earlier researches in the literature that deal with a mainly comparable issue and identify gender discrimination in Bangladesh, the number of studies on women discrimination in Bangladesh is quite low. The current analysis is to uncover potential factors associated with women discrimination based on multilevel logistic analysis.

3. Please improve this sentence “It is observed from the previous study that poor women faced most discrimination compared to wealthy women mostly in every country in the world [18, 22]”

4. The strengths and limitations paragraph need further improvement. The author mentions that the study has several limitations but only mentions one when there are several limitation inherent in this study. Also the strengths of this study are not coming out clearly . For example if there are limited studies that applied multilevel modelling in Bangladesh, how is that a strength?

Conclusion

1. The entire section on conclusion must be reworked. There are lots of hanging statements and repetitions of justifications that have been emphasized in other sections. The conclusion must help the reader understand why your research should matter to them after they have finished reading the paper.

References

1. The referencing format is not presented in an intellectual manner.

Reviewer 02

The manuscript is a technically sound piece of scientific research with data that supports the conclusion. As A reviewer, I suggest author to addd more detailed description in this piece of research as statiscal analysis might not be understood by variety of readers. There are also some sweeping generalized statements which could be modified through standard acadamic English or some more reference should be added to the paragraph of refrence 10-12. Overall quality of the writing needs to be improved.

Reviewers' comments:

Reviewer's Responses to Questions

**Comments to the Author**

1. Is the manuscript technically sound, and do the data support the conclusions?

Reviewer #1: Partly

Reviewer #2: Yes

2. Has the statistical analysis been performed appropriately and rigorously? 

Reviewer #1: No

Reviewer #2: Yes

3. Have the authors made all data underlying the findings in their manuscript fully available?

Reviewer #1: Yes

Reviewer #2: Yes

4. Is the manuscript presented in an intelligible fashion and written in standard English?

Reviewer #1: No

Reviewer #2: No

5. Review Comments to the Author

Reviewer #1: Assessing a multilevel model of women discrimination in Bangladesh: Evidence based

on multiple indicator cluster survey

Article No: PONE-D-23-03956

Corresponding Aurthor : qramul Haq

General comment : The authors are justified to use the approach of statistical analysis opted in this paper, however, the paper is not coherent. There are lots of gaps and standalone statements through out the paper which makes it difficult to appreciate the merits of the paper. There paper requires a professional English editor to correct grammar, accuracy and completeness of some of the statements

Title:

The title of the manuscript is not aligned to the objective of the manuscript. Technically, it is expected that the authors will assess multiple models and eventually report the parsimonious multilevel model, however, this study, sought to investigate the individual and community level factors associated with discrimination among women aged 15-49 years in Bangladesh. Therefore, the author should improve the title accordingly.

Abstract

1.Under Results, change “In the bivariate setup, all the selected covariates except women's education, the ethnicity of the household head, child death, and child sex were found to be significant for discrimination (p <0.05).change statement to read, In the bivariate models, the ethnicity of the household head, child death, and child sex were significantly associated with discrimination.

2. Since the findings included individual and community level factors, they should be reported as such. i.e At the individual level higher odds of discrimination were observed among women from poor (AOR:1.21,95%CI: 1.12-1.32) and middle income households (AOR:1.12 ,95%CI:1.02-1.22) compared to those from richest households etc.

3. The findings should be reported along with their reference group so that it easy for the reader to get complete meaning of the findings on the abstract alone.

4. I suggest the author delete the sentence “Model IV in the multilevel logistic regression model was the best model based on the principles of AIC, BIC, and deviance” instead start the sentence as, Based on the final model (Model1v), at the individual level higher odds of discrimination were observed among women from poor (AOR:1.21,95%CI: 1.12-1.32) and middle income households (AOR:1.12 ,95%CI:1.02-1.22) compared to those from rich households etc

5. Under results there is a statement that reads” Women from the other seven

divisions faced more discrimination than women from the Sylhet division” The actual categories of the variable should be stated along with their results and reference category so that the results are meaningful

Conclusion: The conclusion of the abstract doesn't add much value for this study. “According to the findings of this study, policymakers should focus on individual

and community-level factors that reduce women's discrimination in Bangladesh”. The conclusion should first establish the findings that were found to be significant before the recommendation. Similarly the recommendation is not correct, the research design is cross sectional hence the factors cannot reduce discrimination.

Introduction

1. Even though, I appreciate the background provided by the researcher however in some parts of the writing the author is simply listing statements that are not synthesized carefully to ensure the coherence with the title and the concepts of discrimination. For example, the author was tempted to discuss some of the indicators of discrimination in length such as discrimination on sex, employment, and culture while some were not discussed. It is important that the author first define or discuss what entails discrimination as defined in a standard policy document to focus the study. This will guide the researcher on how to keep the rational of the study coherent with the title and objectives as well as the methodology.

2. A majority of the studies cited are outdated, doesn't provide the prevailing picture of the problem at a global level or local level.

3. Please avoid using statement such as” There is none study conducted in Bangladesh on women discrimination and most of the study applied binary logistic regression “. Instead, I suggest you use, there are limited studies..

3. The rational for the use of a multilevel model as opposed to single level is not clear i.e “That is why we applied a multilevel model to identify both individual and community-level factors that influence women discrimination in Bangladesh. We build this research by examining several socioeconomic, socio-demographic and socio-cultural factors focusing on the general issue of individual perception of women discrimination. We investigate many individual differences that can differential predict reported gender discrimination in women in the present study, as it has been observed that both self-protective and situational factors affect how prejudicial occurrences are interpreted, with a corresponding appropriate methodology”.

Materials and Methods

1. Under the outcome variable subsection, authors should reference Bangladesh MICS to validate the measurement of the outcome variable.

2. Under the covariate subsection it an academic principle to either have a conceptual framework that depict the factors that are possibly associated with the outcome (discrimination) or cite studies that used similar factors to predict discrimination. However, in this study it is difficult to assess if some of the variables used can be used to understand discrimination.

3. Please provide additional information if the author calculated the household wealth index or was already calculated in the dataset. Either way, details on how it was calculated should be provided.

4. Under statistical analysis the authors make no mention of descriptive statistics, yet it is expected the prevalence of discrimination will be reported as well as the sample distribution. Please improve the section accordingly.

5. Even though the structure of the dataset is accurately described by the authors , most of the justification remain an opinion, no literature is cited to rationalize the use of the multilevel model i.e paragraph one under statistical analysis states “

It should be emphasized that a two-stage stratified sampling approach was used to get the data for this investigation. As a result, there is a hierarchy of levels at which reliance between observations occurs. The data set has a community-level influence. Instead of using a single-level modelling strategy, one could think of using multilevel modeling to examine this kind of data”.

6. Under Statistical analysis only Bivariate and multivariate analyses techniques are cited as approaches for the analysis in this study. However, there are results on sample distribution (see table 1) which are are not accommodated by the two analysis approaches. Please improve accordingly.

Results

1. Table 1: It may be good to list the variables according to the classification i.e list individual variables first and then community level.

2. The author may consider making the table in excel for good presentation.

3. You may consider having sub sections under the results

4. Just before the results of table 2, there should be a sub section: the relationship between discrimination and individual and community variables

5. There are missing results of the overall prevalence of discrimination that must be presented along with confidence intervals

6. The results in table 2 on the association between selected covariates and discrimination against women in Bangladesh are interpreted incorrectly. The results are row percent yet they are interpreted as column percent which is not correct.

7. The Wealth index is a proxy of the standard of living for households not the women, that is why household amenities are used to derive the household wealth index. Therefore improve the statement/interpretation of the results for the multilevel model, (table 3). Please add “The household wealth index showed a significant relationship with discrimination. The results demonstrated that women from less wealth households were more likely to face discrimination than women from rich households. For example, poor women from poor households were 21% (OR = 1.21, 95% CI: 1.12-1.32) more likely discriminated than women from rich households”.

8. Generally some interpretation requires editing for grammar to ensure they make sense to the reader. The entire section need to be improved. Some of the glaring interpretations:

• The odds of discrimination were 94% higher (OR=1.94, 95%, CI: 1.69-2.22) in women who has no child than in woman with 3+ child.

• Women not exposed to information and communication network were 27% more (OR = 1.27, 95% CI=1.07-1.51) discriminated than those were exposed.

• Women currently married were 49% less (OR = 0:51, 95% CI: 0.44-0.59), and women formerly married were 23% less (OR = 0.77, 95% CI: 0.65-0.92) faced discrimination than the reference category women never married.

• The rate of discrimination was significantly 13% (OR = 1.13, 95% CI: 1.04-1.23) more for 20-34 years old women compare to those were 35-49 years old. This is not a rate instead ratios. So technically the interpretation is not correct

• According to model IV in Table 3, happy women were 59% (OR = 0.41, 95% CI: 0.36-0.46) less likely to face discrimination than those who were unhappy.

• Women who did not read magazines were 17% less likely to be discriminated (OR = 0.83, 95% CI 0.74-0.92) less likely to be discriminated than those who did not read magazines.

9. Instead of using OR for model 3 for the fixed effects, use AOR

10. Please provide the PVC-proportion change in variance for each model.

11. Please format table 3.

Discussion

1. The Discussion section need major attention. The author was just listing the factors that were found to be associated with discrimination without a detailed synthesis of the findings. Moreover, The discussion chapter mostly identifies how findings are supported or support previous studies. I suggest the author also include studies that deviate from the extant literature.

2. I suggest you delete the first paragraph under Discussion “This study collects the necessary information from the secondary data set named 2019 multiple indicator cluster (MICS) in Bangladesh. The multilevel logistic regression model is used in this study. Although there are some earlier researches in the literature that deal with a mainly comparable issue and identify gender discrimination in Bangladesh, the number of studies on women discrimination in Bangladesh is quite low. The current analysis is to uncover potential factors associated with women discrimination based on multilevel logistic analysis.

3. Please improve this sentence “It is observed from the previous study that poor women faced most discrimination compared to wealthy women mostly in every country in the world [18, 22]”

4. The strengths and limitations paragraph need further improvement. The author mentions that the study has several limitations but only mentions one when there are several limitation inherent in this study. Also the strengths of this study are not coming out clearly . For example if there are limited studies that applied multilevel modelling in Bangladesh, how is that a strength?

Conclusion

1. The entire section on conclusion must be reworked. There are lots of hanging statements and repetitions of justifications that have been emphasized in other sections. The conclusion must help the reader understand why your research should matter to them after they have finished reading the paper.

References

1. The referencing format is not presented in an intellectual manner.

Reviewer #2: The manuscript is a technically sound piece of scientific research with data that supports the conclusion. As A reviewer, I suggest author to addd more detailed description in this piece of research as statiscal analysis might not be understood by variety of readers. There are also some sweeping generalized statements which could be modified through standard acadamic English or some more reference should be added to the paragraph of refrence 10-12. Overall quality of the writing needs to be improved.

6. PLOS authors have the option to publish the peer review history of their article (what does this mean?). If published, this will include your full peer review and any attached files.

Reviewer #1: No

Reviewer #2: **Yes: **Dr. Zubaida Zafar

---

## [Author Response · Author response to Decision Letter 0]

30 Jun 2023

Dear Editor,

On behalf of all the authors, I wish to convey our gratitude to you for the critical and constructive review that has led to the improvement of our manuscript entitled “Individual and Community-level factors associated with discrimination among women aged 15-49 years in Bangladesh: Evidence based on multiple indicator cluster survey.” We have revised the manuscript based on the comments raised by the reviewers. Along with reviewers comments we also revised manuscript and improved its English level. We believe the manuscript has improved according to reviewers comments and its English level substantively and will be published in your reputable Journal, “PLOS ONE.” All the changes are highlighted in the yellow color in the revised manuscript. I hope our efforts satisfy the requirements of the journal this time. I will be looking forward to your positive response. Thanks, and regards.

On the behalf of authors

Iqramul Haq

Corresponding author

PONE-D-23-03956

Assessing a multilevel model of women discrimination in Bangladesh: Evidence based on multiple indicator cluster survey

PLOS ONE

Dear Dr. Haq,

Thank you for submitting your manuscript to PLOS ONE. After careful consideration, we feel that it has merit but does not fully meet PLOS ONE’s publication criteria as it currently stands. Therefore, we invite you to submit a revised version of the manuscript that addresses the points raised during the review process.

Author Response: We would like to express our gratitude for your positive response.

Journal Requirements:

Author Response: Checked.

Author Response: Thank you for advice. We made necessary change of the revised manuscript.

Author Response: Checked.

Following changes are recommenced from reviewers that need to be incorporated:

## The following is a point-by-point response to the reviewer(s) comments:

#Reviewer 01:

General comment : The authors are justified to use the approach of statistical analysis opted in this paper, however, the paper is not coherent. There are lots of gaps and standalone statements through out the paper which makes it difficult to appreciate the merits of the paper. There paper requires a professional English editor to correct grammar, accuracy and completeness of some of the statements

Author Response: We would like to express our gratitude for your positive response. In light of your comments, we have made revisions and enhancements to improve aspects of the English language, such as grammar, accuracy, and completeness of certain statements.

Title:

The title of the manuscript is not aligned to the objective of the manuscript. Technically, it is expected that the authors will assess multiple models and eventually report the parsimonious multilevel model, however, this study, sought to investigate the individual and community level factors associated with discrimination among women aged 15-49 years in Bangladesh. Therefore, the author should improve the title accordingly.

Author Response: Based on the comments received, we have made revisions to the manuscript's title.

Abstract

1.Under Results, change “In the bivariate setup, all the selected covariates except women's education, the ethnicity of the household head, child death, and child sex were found to be significant for discrimination (p <0.05).change statement to read, In the bivariate models, the ethnicity of the household head, child death, and child sex were significantly associated with discrimination.

Author Response: Thank you for informing us. We have incorporated the necessary changes in response to the comments received.

2. Since the findings included individual and community level factors, they should be reported as such. i.e At the individual level higher odds of discrimination were observed among women from poor (AOR:1.21,95%CI: 1.12-1.32) and middle income households (AOR:1.12 ,95%CI:1.02-1.22) compared to those from richest households etc.

Author Response: Thank you for your suggestions. We have revised the result section of the abstract and incorporated the necessary additions.

3. The findings should be reported along with their reference group so that it easy for the reader to get complete meaning of the findings on the abstract alone.

Author Response: We have made the necessary changes to the manuscript as required.

4. I suggest the author delete the sentence “Model IV in the multilevel logistic regression model was the best model based on the principles of AIC, BIC, and deviance” instead start the sentence as, Based on the final model (Model1v), at the individual level higher odds of discrimination were observed among women from poor (AOR:1.21,95%CI: 1.12-1.32) and middle income households (AOR:1.12 ,95%CI:1.02-1.22) compared to those from rich households etc

Author Response: We appreciate your suggestions. We have made the necessary revisions and included the recommended additions in the manuscript.

5. Under results there is a statement that reads” Women from the other seven

divisions faced more discrimination than women from the Sylhet division” The actual categories of the variable should be stated along with their results and reference category so that the results are meaningful

Author Response: Thank you for your suggestions. We have made revisions to the manuscript in accordance with your suggestion.

Conclusion: The conclusion of the abstract doesn't add much value for this study. “According to the findings of this study, policymakers should focus on individual

and community-level factors that reduce women's discrimination in Bangladesh”. The conclusion should first establish the findings that were found to be significant before the recommendation. Similarly the recommendation is not correct, the research design is cross sectional hence the factors cannot reduce discrimination.

Author Response: Thank you for your suggestions. Revised with additions to the conclusion section of the abstract's sub-section.

Introduction

1. Even though, I appreciate the background provided by the researcher however in some parts of the writing the author is simply listing statements that are not synthesized carefully to ensure the coherence with the title and the concepts of discrimination. For example, the author was tempted to discuss some of the indicators of discrimination in length such as discrimination on sex, employment, and culture while some were not discussed. It is important that the author first define or discuss what entails discrimination as defined in a standard policy document to focus the study. This will guide the researcher on how to keep the rational of the study coherent with the title and objectives as well as the methodology.

Author Response: Added in the revised manuscript.

2. A majority of the studies cited are outdated, doesn't provide the prevailing picture of the problem at a global level or local level.

Author Response: Revised.

3. Please avoid using statement such as” There is none study conducted in Bangladesh on women discrimination and most of the study applied binary logistic regression “. Instead, I suggest you use, there are limited studies..

Author Response: Revised and checked. 

3. The rational for the use of a multilevel model as opposed to single level is not clear i.e “That is why we applied a multilevel model to identify both individual and community-level factors that influence women discrimination in Bangladesh. We build this research by examining several socioeconomic, socio-demographic and socio-cultural factors focusing on the general issue of individual perception of women discrimination. We investigate many individual differences that can differential predict reported gender discrimination in women in the present study, as it has been observed that both self-protective and situational factors affect how prejudicial occurrences are interpreted, with a corresponding appropriate methodology”.

Author Response: We have made the necessary changes to the revised manuscript.

Materials and Methods

1. Under the outcome variable subsection, authors should reference Bangladesh MICS to validate the measurement of the outcome variable.

Author Response: We have added the citation under the subsection of the outcome variable, as you suggested.

2. Under the covariate subsection it an academic principle to either have a conceptual framework that depict the factors that are possibly associated with the outcome (discrimination) or cite studies that used similar factors to predict discrimination. However, in this study it is difficult to assess if some of the variables used can be used to understand discrimination.

Author Response: We have added a conceptual framework under the covariate subsection of the revised manuscript.

3. Please provide additional information if the author calculated the household wealth index or was already calculated in the dataset. Either way, details on how it was calculated should be provided.

Author Response: Thank you. We have included the relevant information in this section of the revised manuscript.

4. Under statistical analysis the authors make no mention of descriptive statistics, yet it is expected the prevalence of discrimination will be reported as well as the sample distribution. Please improve the section accordingly.

Author Response: We have incorporated the necessary revisions into our revised manuscript.

5. Even though the structure of the dataset is accurately described by the authors , most of the justification remain an opinion, no literature is cited to rationalize the use of the multilevel model i.e paragraph one under statistical analysis states “

It should be emphasized that a two-stage stratified sampling approach was used to get the data for this investigation. As a result, there is a hierarchy of levels at which reliance between observations occurs. The data set has a community-level influence. Instead of using a single-level modelling strategy, one could think of using multilevel modeling to examine this kind of data”.

Author Response: We have added to our revised manuscript based on your suggestions.

6. Under Statistical analysis only Bivariate and multivariate analyses techniques are cited as approaches for the analysis in this study. However, there are results on sample distribution (see table 1) which are are not accommodated by the two analysis approaches. Please improve accordingly.

Author Response: We have included the necessary additions in the revised manuscript.

Results

1. Table 1: It may be good to list the variables according to the classification i.e list individual variables first and then community level.

Author Response: Checked and corrected. 

2. The author may consider making the table in excel for good presentation.

Author Response: Thank you for your suggestions. We have make necessary change of the manuscript.

3. You may consider having sub sections under the results

Author Response: Thank you for your comments. We have added a subsection under the result section of this manuscript.

4. Just before the results of table 2, there should be a sub section: the relationship between discrimination and individual and community variables

Author Response: We have added in subsection under the result section of this revised manuscript.

5. There are missing results of the overall prevalence of discrimination that must be presented along with confidence intervals

Author Response: We have added the overall prevalence of discrimination along with confidence intervals to this manuscript.

6. The results in table 2 on the association between selected covariates and discrimination against women in Bangladesh are interpreted incorrectly. The results are row percent yet they are interpreted as column percent which is not correct.

Author Response: We have made the necessary corrections in our revised manuscript.

7. The Wealth index is a proxy of the standard of living for households not the women, that is why household amenities are used to derive the household wealth index. Therefore improve the statement/interpretation of the results for the multilevel model, (table 3). Please add “The household wealth index showed a significant relationship with discrimination. The results demonstrated that women from less wealth households were more likely to face discrimination than women from rich households. For example, poor women from poor households were 21% (OR = 1.21, 95% CI: 1.12-1.32) more likely discriminated than women from rich households”.

Author Response: Thank you for your advice. We have made the necessary changes to the revised manuscript.

8. Generally some interpretation requires editing for grammar to ensure they make sense to the reader. The entire section need to be improved. Some of the glaring interpretations:

• The odds of discrimination were 94% higher (OR=1.94, 95%, CI: 1.69-2.22) in women who has no child than in woman with 3+ child.

Author Response: Thank you for your advice. Thank you for your advice. We have added to the revised manuscript.

• Women not exposed to information and communication network were 27% more (OR = 1.27, 95% CI=1.07-1.51) discriminated than those were exposed.

Author Response: Thank you for your advice. We have included the necessary additions in the revised manuscript.

• Women currently married were 49% less (OR = 0:51, 95% CI: 0.44-0.59), and women formerly married were 23% less (OR = 0.77, 95% CI: 0.65-0.92) faced discrimination than the reference category women never married.

Author Response: We have added to the revised manuscript.

• The rate of discrimination was significantly 13% (OR = 1.13, 95% CI: 1.04-1.23) more for 20-34 years old women compare to those were 35-49 years old. This is not a rate instead ratios. So technically the interpretation is not correct

Author Response: We have added to the revised manuscript.

• According to model IV in Table 3, happy women were 59% (OR = 0.41, 95% CI: 0.36-0.46) less likely to face discrimination than those who were unhappy.

Author Response: We have added to the revised manuscript.

• Women who did not read magazines were 17% less likely to be discriminated (OR = 0.83, 95% CI 0.74-0.92) less likely to be discriminated than those who did not read magazines.

Author Response: We have added to the revised manuscript.

9. Instead of using OR for model 3 for the fixed effects, use AOR

Author Response: Thank you for your advice. We have added to the revised manuscript.

10. Please provide the PVC-proportion change in variance for each model.

Author Response: Thank you for your advice. We have made necessary change of the revised manuscript.

11. Please format table 3. 

Author Response: Revised

Discussion

1. The Discussion section need major attention. The author was just listing the factors that were found to be associated with discrimination without a detailed synthesis of the findings. Moreover, The discussion chapter mostly identifies how findings are supported or support previous studies. I suggest the author also include studies that deviate from the extant literature.

Author Response: Thank you for your advice. We have made necessary change of the revised manuscript.

2. I suggest you delete the first paragraph under Discussion “This study collects the necessary information from the secondary data set named 2019 multiple indicator cluster (MICS) in Bangladesh. The multilevel logistic regression model is used in this study. Although there are some earlier researches in the literature that deal with a mainly comparable issue and identify gender discrimination in Bangladesh, the number of studies on women discrimination in Bangladesh is quite low. The current analysis is to uncover potential factors associated with women discrimination based on multilevel logistic analysis.

Author Response: Checked and corrected.

3. Please improve this sentence “It is observed from the previous study that poor women faced most discrimination compared to wealthy women mostly in every country in the world [18, 22]”

Author Response: Revised and corrected.

4. The strengths and limitations paragraph need further improvement. The author mentions that the study has several limitations but only mentions one when there are several limitation inherent in this study. Also the strengths of this study are not coming out clearly . For example if there are limited studies that applied multilevel modelling in Bangladesh, how is that a strength?

Author Response: Thank you for your advice. We have made necessary change of the revised manuscript.

Conclusion

1. The entire section on conclusion must be reworked. There are lots of hanging statements and repetitions of justifications that have been emphasized in other sections. The conclusion must help the reader understand why your research should matter to them after they have finished reading the paper.

Author Response: Thank you for your advice. We have made necessary change of the revised manuscript.

References

1. The referencing format is not presented in an intellectual manner.

Author Response: Revised and checked.

#Reviewer 02

The manuscript is a technically sound piece of scientific research with data that supports the conclusion. As A reviewer, I suggest author to addd more detailed description in this piece of research as statiscal analysis might not be understood by variety of readers. There are also some sweeping generalized statements which could be modified through standard acadamic English or some more reference should be added to the paragraph of refrence 10-12. Overall quality of the writing needs to be improved.

Author Response: Thank you for your suggestions. We have made necessary change of the revised manuscript.

##I hope our efforts satisfy the requirements of the journal this time. I will be looking forward to your positive response. Thanks, and regards.

---

## [Editor Report · Decision Letter 1]

10 Jul 2023

Individual and Community-level factors associated with discrimination among women aged 15-49 years in Bangladesh: Evidence based on multiple indicator cluster survey

PONE-D-23-03956R1

Dear Dr. Haq,

We’re pleased to inform you that your manuscript has been judged scientifically suitable for publication and will be formally accepted for publication once it meets all outstanding technical requirements.

Kind regards,

Sadia Jabeen, Ph.D.

Academic Editor

PLOS ONE
---

## [Editor Report · Acceptance letter]

18 Jul 2023

PONE-D-23-03956R1 

Individual and Community-level factors associated with discrimination among women aged 15-49 years in Bangladesh: Evidence based on multiple indicator cluster survey 

Dear Dr. Haq:

I'm pleased to inform you that your manuscript has been deemed suitable for publication in PLOS ONE. Congratulations! Your manuscript is now with our production department. 

Kind regards, 

on behalf of

Dr. Sadia Jabeen 

Academic Editor

PLOS ONE